# Quadratic Metric Elicitation for Fairness and Beyond

**Gaurush Hiranandani**[1]     **Jatin Mathur**[2]     **Harikrishna Narasimhan**[3]     **Oluwasanmi Koyejo**[2,3]

[1]Amazon, Palo Alto, California, USA
[2]University of Illinois at Urbana-Champaign, Champaign, Illinois, USA
[3]Google Research, Mountain View, California, USA

## Abstract

Metric elicitation is a recent framework for eliciting classification performance metrics that best reflect implicit user preferences based on the task and context. However, available elicitation strategies have been limited to linear (or quasi-linear) functions of predictive rates, which can be practically restrictive for many applications including fairness. This paper develops a strategy for eliciting more flexible multiclass metrics defined by quadratic functions of rates, designed to reflect human preferences better. We show its application in eliciting quadratic violation-based group-fair metrics. Our strategy requires only relative preference feedback, is robust to noise, and achieves near-optimal query complexity. We further extend this strategy to eliciting polynomial metrics – thus broadening the use cases for metric elicitation.

## 1   INTRODUCTION

*Given a classification task, which performance metric should the classifier optimize?* This question is often faced by practitioners while developing machine learning solutions. For example, consider cancer diagnosis where a doctor applies a cost-sensitive predictive model to classify patients into cancer categories [Yang and Naiman, 2014]. The costs may be based on known consequences of misdiagnosis, i.e, side-effects of treating a healthy patient vs. mortality rate for not treating a sick patient. Although it is clear that the chosen costs directly determine the model decisions and dictate patient outcomes, it is not clear how to quantify the expert's intuition into precise quantitative cost trade-offs, i.e., the performance metric.

Indeed, the above is also true for a variety of other domains including *fair machine learning* where picking the right metric is a critical challenge [Dmitriev and Wu, 2016, Zhang

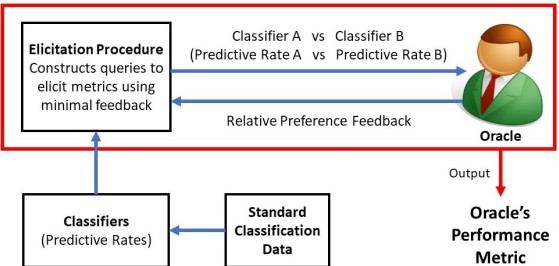

Figure 1: Metric Elicitation [Hiranandani et al., 2019a].

et al., 2020]. The issue is exacerbated when the practitioner's notion of fairness does not exactly match with any standard fairness criterion. For example, a practitioner may be interested in weighting each group discrepancy differently, but may not be able to provide us with the exact weights or a precise mathematical expression that reflects on the practitioner's innate fairness notion.

Hiranandani et al. [2019a,b, 2020] addressed this issue by formalizing the framework of *Metric Elicitation (ME)*, whose goal is to estimate a performance metric using preference feedback from a user. The motivation is that by employing metrics that reflect a user's innate trade-offs given the task, context, and population at hand, one can learn models that best capture the user preferences [Hiranandani et al., 2019a]. As humans are often inaccurate in providing absolute quality feedback [Qian et al., 2013], Hiranandani et al. [2019a] propose to use pairwise comparison queries, where the user (oracle) is asked to compare two classifiers and provide a relative preference. Using such pairwise comparison queries, ME aims to recover the oracle's metric. Figure 1 (reproduced from Hiranandani et al. [2019a]) depicts the ME framework.

A notable drawback of existing ME strategies is that they only handle linear or quasi-linear function of predictive rates, which can be restrictive for many applications where the metrics are non-linear. For example, in *fair machine learning*, classifiers are often judged by measuring discrepancies between predictive rates for different protected groups [Hardt et al., 2016]. Similarly, discrepancies among

*Accepted for the 38th Conference on Uncertainty in Artificial Intelligence* (UAI 2022).

different distributions are measured in *distribution matching* applications [Narasimhan, 2018, Esuli and Sebastiani, 2015]. A common measure of discrepancy in such applications is the squared difference, which is a quadratic metric that cannot be handled by existing approaches. Quadratic metrics also find use in class-imbalanced learning [Goh et al., 2016, Narasimhan, 2018] (see Section 2.3 for examples). Motivated by these examples, in this paper, we propose strategies for eliciting metrics defined by *quadratic* functions of rates, that encompass linear metrics as special cases. Our approach also generalizes to eliciting polynomial metrics, a universal family of functions [Stone, 1948], allowing one to better capture real-world human preferences.

Our high-level idea is to approximate the quadratic metric with multiple linear functions, employ linear ME to estimate the individual local slopes, and combine the slope estimates to reconstruct the original metric. While natural and elegant, this approach comes with non-trivial challenges. Firstly, we must choose center points for the local-linear approximations, and the chosen points must represent feasible queries. Secondly, because of the use of pairwise queries, we only receive *slopes* (directions) and not magnitudes for the local-linear functions, requiring intricate analyses to reconstruct the original metric and to deal with multiplicative errors that result. Despite the challenges, our method requires a query complexity that is only *linear* in the number of unknowns, which we show is *near-optimal*. To our knowledge, we are the first to prove such a lower bound for metric elicitation.

We further elaborate on eliciting group-fair metrics. The prior work by Hiranandani et al. [2020] consider a restricted class of fairness metrics, where the fairness discrepancies are defined to be the *absolute* differences between group-specific rates. Moreover, their approach does not generalize to other families of metrics. In contrast, we are able to handle a more general family of non-linear fairness metrics defined by quadratic functions of group rate differences and show how our proposed quadratic ME approach is easily adaptable to elicit such group-fair quadratic metrics.

In summary, we make the following contributions :

- We propose a novel quadratic ME algorithm for classification problems, which requires only pairwise preference feedback either over classifiers or predictive rates.
- Specific to group-based fairness tasks, we show how to jointly elicit the predictive performance and fairness metrics, and the trade-off between them.
- We show that the proposed approach is robust under feedback and finite sample noise and requires a near-optimal number of queries.
- We empirically validate the proposal for multiple classes and groups on simulated oracles.
- We discuss how our strategy can be generalized to elicit higher-order polynomials by recursively applying the procedure to elicit lower-order approximations.

**Paper Organization:** For ease of exposition, we first discuss quadratic metric elicitation in the usual multiclass classification setup without fairness. Section 2 contains the problem setup and the associated background, and Section 3 describes the proposed quadratic ME procedure. We then cover ME under the multiclass-multigroup framework in Section 4, where we additionally have protected group information embedded in the problem setup. In Section 5, we provide guarantees for our proposed procedures, and in Section 6, we present our experiments. We discuss related work in Section 7 and provide concluding remarks in Section 8.

**Notations.** For $k \in \mathbb{Z}_+$, we denote $[k] = \{1, \cdots, k\}$ and use $\Delta_k$ to denote the $(k-1)$-dimensional simplex. We denote inner products by $\langle \cdot, \cdot \rangle$ and Hadamard products by $\odot$. $\| \cdot \|_F$ represents the Frobenius norm, and $\boldsymbol{\alpha}_i \in \mathbb{R}^q$ denotes the $i$-th standard basis vector, where the $i$-th coordinate is 1 and others are 0.

## 2 BACKGROUND

We consider a $k$-class classification setting with $X \in \mathcal{X}$ and $Y \in [k]$ denoting the input and output random variables, respectively. We assume access to an $n$-sized sample $\{(\mathbf{x}, y)_i\}_{i=1}^n$ generated *iid* from a distribution $\mathbb{P}(X, Y)$. We work with randomized classifiers $h : \mathcal{X} \to \Delta_k$ that for any $\mathbf{x}$ gives a distribution $h(\mathbf{x})$ over the $k$ classes and use $\mathcal{H} = \{h : \mathcal{X} \to \Delta_k\}$ to denote the set of all classifiers.

*Predictive rates:* We denote the predictive rates for a classifier $h$ by the vector $\mathbf{r}(h, \mathbb{P}) \in \mathbb{R}^k$, where the $i$-th coordinate is the fraction of label-$i$ examples for which the randomized classifier $h$ also predicts class $i$:

$$r_i(h, \mathbb{P}) := \mathbb{P}(h(X) = i | Y = i) \quad \text{for } i \in [k]. \quad (1)$$

The probability above is over draw of $(X, Y) \sim \mathbb{P}$ and the randomness in $h$. The proposed setup and solution (discussed later) easily extends to general predictive rates of the form $\mathbb{P}(h(X) = j | Y = i)$ for $i, j \in [k]$. For simplicity, we defer this extension to Appendix E.

*Metrics:* We consider metrics that are defined by a general function $\phi : [0, 1]^k \to \mathbb{R}$ of rates:

$$\phi(\mathbf{r}(h, \mathbb{P})).$$

This includes the (weighted) accuracy $\phi^{\text{acc}}(\mathbf{r}(h, \mathbb{P})) = \sum_i a_i r_i(h, \mathbb{P})$, for weights $a_i \in \mathbb{R}_+$, the G-mean, and many more metrics [Sokolova and Lapalme, 2009]. Unless specified, we treat metrics as utilities, i.e., larger values are better. Since the metric's scale does not affect the learning problem [Narasimhan et al., 2015], we allow $\phi : [0, 1]^k \to [0, 1]$.

*Feasible rates:* We will restrict our attention to only those rates that are feasible, i.e., can be achieved by some classifier. The set of all feasible rates is given by:

$$\mathcal{R} = \{\mathbf{r}(h, \mathbb{P}) : h \in \mathcal{H}\}.$$

To avoid clutter in notations, we will suppress the dependence on $\mathbb{P}$ and $h$ if it is clear from the context.

## 2.1 METRIC ELICITATION: PROBLEM SETUP

We now describe the problem of *Metric Elicitation*, which follows from Hiranandani et al. [2019b]. There's an *unknown* metric $\phi$, and we seek to elicit its form by posing queries to an *oracle* asking which of two classifiers is more preferred by it. The oracle has access to the metric $\phi$ and responds by comparing its value on the two classifiers.

**Definition 1** (Oracle Query). *Given two classifiers $h_1, h_2$ (equiv. to rates $\mathbf{r}_1, \mathbf{r}_2$ respectively), a query to the Oracle (with metric $\phi$) is represented by:*

$$\Gamma(h_1, h_2 \,;\, \phi) = \Omega(\mathbf{r}_1, \mathbf{r}_2 \,;\, \phi) = \mathbb{1}[\phi(\mathbf{r}_1) > \phi(\mathbf{r}_2)], \quad (2)$$

*where $\Gamma : \mathcal{H} \times \mathcal{H} \to \{0, 1\}$ and $\Omega : \mathcal{R} \times \mathcal{R} \to \{0, 1\}$. The query asks whether $h_1$ is preferred to $h_2$ (equiv. if $\mathbf{r}_1$ is preferred to $\mathbf{r}_2$), as measured by $\phi$.*

In practice, the oracle can be an expert, a group of experts, or an entire user population. The ME framework can be applied by posing classifier comparisons directly via interpretable learning techniques [Ribeiro et al., 2016] or via A/B testing [Tamburrelli and Margara, 2014]. For example, in an internet-based application one may perform the A/B test by deploying two classifiers A and B with two different sub-populations of users and use their level of engagement to decide the preference over the two classifiers. For other applications, one may present visualizations of rates of the two classifiers (e.g., [Shen et al., 2020]), and have the user provide the preference (see Appendix J for an example). Moreover, since the metrics we consider are functions of only the predictive rates, queries comparing classifiers are the same as queries on the associated rates. So for convenience, we will have our algorithms pose queries comparing two (feasible) rates. Indeed given a feasible rate, one can efficiently find the associated classifier (see Appendix B.1 for details). We next formally state the ME problem.

**Definition 2** (Metric Elicitation with Pairwise Queries (given $\{(\mathbf{x}, y)_i\}_{i=1}^n$) [Hiranandani et al., 2019a,b]). *Suppose that the oracle's (unknown) performance metric is $\phi$. Using oracle queries of the form $\Omega(\hat{\mathbf{r}}_1, \hat{\mathbf{r}}_2 \,;\, \phi)$, where $\hat{\mathbf{r}}_1, \hat{\mathbf{r}}_2$ are the estimated rates from samples, recover a metric $\hat{\phi}$ such that $\|\phi - \hat{\phi}\| < \kappa$ under a suitable norm $\|\cdot\|$ for sufficiently small error tolerance $\kappa > 0$.*

The performance of ME is evaluated both by the query complexity and the quality of the elicited metric [Hiranandani et al., 2019a,b]. As is standard in the decision theory literature [Koyejo et al., 2015], we present our ME approach by first assuming access to population quantities such as the population rates $\mathbf{r}(h, \mathbb{P})$, then examine estimation error from finite samples, i.e., with empirical rates $\hat{\mathbf{r}}(h, \{(\mathbf{x}, y)_i\}_{i=1}^n)$.

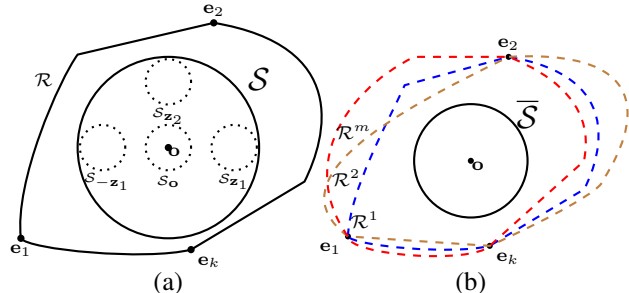

Figure 2: (a) Geometry of the set of predictive rates $\mathcal{R}$: A convex set enclosing a sphere $\mathcal{S}$ with trivial rates $\mathbf{e}_i \, \forall \, i \in [k]$ as vertices; (b) Geometry of the product set of group rates $\mathcal{R}^1 \times \cdots \times \mathcal{R}^m$ (best seen in color) [Hiranandani et al., 2020]; $\mathcal{R}^u \, \forall \, u \in [m]$ are convex sets with common vertices $\mathbf{e}_i \, \forall \, i \in [k]$ and enclose a sphere $\overline{\mathcal{S}} \subset \mathcal{R}^1 \cap \cdots \cap \mathcal{R}^m$.

## 2.2 LINEAR METRIC ELICITATION

As a warm up, we overview the Linear Performance Metric Elicitation (LPME) procedure of [Hiranandani et al., 2019b], which we will use as a subroutine. Here we assume that the oracle's metric is a linear function of rates $\phi^{\text{lin}}(\mathbf{r}) := \langle \mathbf{a}, \mathbf{r} \rangle$, for some unknown weights $\mathbf{a} \in \mathbb{R}^k$. In other words, given two rates $\mathbf{r}_1$ and $\mathbf{r}_2$, the oracle returns $\mathbb{1}[\langle \mathbf{a}, \mathbf{r}_1 \rangle > \langle \mathbf{a}, \mathbf{r}_2 \rangle]$. Since the metrics are scale invariant [Narasimhan et al., 2015], without loss of generality (w.l.o.g.), one may assume $\|\mathbf{a}\|_2 = 1$. The goal is to elicit (the slope of) $\mathbf{a}$ using pairwise comparisons over rates.

When the number of classes $k = 2$, the coefficients $\mathbf{a}$ can be elicited using a one-dimensional binary search. When $k > 2$, one can apply a coordinate-wise procedure, performing a binary search in one coordinate, while keeping the others fixed. The efficacy of this procedure, however, hinges on the geometry of the set of rates $\mathcal{R}$. Before discussing the geometry, we make a mild assumption that ensures some signal for non-trivial classification.

**Assumption 1.** *The conditional-class distributions are distinct, i.e., $\mathbb{P}(Y = i|X) \neq \mathbb{P}(Y = j|X) \, \forall \, i \neq j$.*

Let $\mathbf{e}_i \in \{0, 1\}^k$ denote the rates achieved by a trivial classifier that predicts class $i$ for all inputs.

**Proposition 1** (Geometry of $\mathcal{R}$; Figure 2(a)). *The set of rates $\mathcal{R} \subseteq [0, 1]^k$ is convex, has vertices $\{\mathbf{e}_i\}_{i=1}^k$, and contains the rate profile $\mathbf{o} = \frac{1}{k}\sum_{i=1}^k \mathbf{e}_i$ in the interior. Moreover, $\mathbf{o}$ is achieved by the uniform random classifier which for any input predicts each class with equal probability.*

**Remark 1** (Existence of sphere $\mathcal{S}$). *Since $\mathcal{R}$ is convex and contains the point $\mathbf{o}$ in the interior, there exists a sphere $\mathcal{S} \subset \mathcal{R}$ of non-zero radius $\rho$ centered at $\mathbf{o}$.*

By restricting the coordinate-wise binary search procedure to posing queries from within a sphere, LPME can be seen

as minimizing a strongly-convex function and shown to converge to a solution $\hat{\mathbf{a}}$ close to $\mathbf{a}$. Specifically, the LPME procedure takes any sphere $\mathcal{S} \subset \mathcal{R}$, binary-search tolerance $\epsilon$, and the oracle $\Omega$ (with metric $\phi^{\text{lin}}$) as input, and by posing $O(k \log(1/\epsilon))$ queries recovers coefficients $\hat{\mathbf{a}}$ with $\|\mathbf{a} - \hat{\mathbf{a}}\|_2 \leq O(\sqrt{k}\epsilon)$. The details of the algorithm are provided in Appendix A for completeness, but the following remark is the most important for our subsequent discussion.

**Remark 2** (LPME Guarantee). *Given any $k$-dimensional sphere $\mathcal{S} \subset \mathcal{R}$ and an oracle $\Omega$ with metric $\phi^{\text{lin}}(\mathbf{r}) := \langle \mathbf{a}, \mathbf{r} \rangle$, the LPME algorithm (Algorithm 2, Appendix A) provides an estimate $\hat{\mathbf{a}}$ with $\|\hat{\mathbf{a}}\|_2 = 1$ such that the estimated slope is close to the true slope, i.e., $a_i/a_j \approx \hat{a}_i/\hat{a}_j \; \forall \; i, j \in [k]$.*

Note that the LPME procedure is closely tied to the scale invariance condition and thus only estimates the slope (direction) of the coefficient vector $\mathbf{a}$, and not its magnitude. Despite this drawback, we will discuss how we can elicit quadratic metrics using LPME in Section 3. Also note the algorithm takes as input an *arbitrary* sphere $\mathcal{S} \subset \mathcal{R}$, and restricts its queries to rate vectors within the sphere. In Appendix B.1, we discuss an efficient procedure [Hiranandani et al., 2019b] for identifying a sphere of suitable radius.

### 2.3 QUADRATIC PERFORMANCE METRICS

Equipped with the LPME subroutine, our aim is to elicit metrics that are quadratic functions of rates.

**Definition 3** (Quadratic Metric). *For a vector $\mathbf{a} \in \mathbb{R}^k$ and a negative semi-definite matrix $\mathbf{B} \in NSD_k$ with $\|\mathbf{a}\|_2^2 + \|\mathbf{B}\|_F^2 = 1$ (w.l.o.g. due to scale invariance):*

$$\phi^{\text{quad}}(\mathbf{r}\,;\, \mathbf{a}, \mathbf{B}) = \langle \mathbf{a}, \mathbf{r} \rangle + \frac{1}{2}\mathbf{r}^T \mathbf{B} \mathbf{r}. \quad (3)$$

This family trivially includes the linear metrics as well as many modern metrics outlined below:

**Example 1** (Class-imbalanced learning). In problems with imbalanced class proportions, it is common to use metrics that emphasize equal performance across all classes. One example is Q-mean [Menon et al., 2013], which is the quadratic mean of rates: $\phi^{\text{qmean}}(\mathbf{r}) = 1 - 1/k \sum_{i=1}^{k} (1 - r_i)^2$.

**Example 2** (Distribution matching). In certain binary classification applications, one needs the proportion of predictions for each class (i.e., the coverage) to match a target distribution $\boldsymbol{\pi} \in \Delta_2$ [Goh et al., 2016, Narasimhan, 2018]. A measure often used for this task is the squared difference between the per-class coverage and the target distribution: $\phi^{\text{cov}}(\mathbf{r}) = 1 - \frac{1}{2} \sum_{i=1}^{2} (\text{cov}_i(\mathbf{r}) - \pi_i)^2$, where $\text{cov}_i(\mathbf{r}) = r_i + 1 - r_{\neq i}$. Similar metrics can be found in the quantification literature where the target is set to the class prior $\mathbb{P}(Y = i)$ [Esuli and Sebastiani, 2015, Kar et al., 2016]. We capture more general quadratic distance measures for distributions, e.g. $(\text{cov}(\mathbf{r}) - \boldsymbol{\pi})^T \mathbf{Q}(\text{cov}(\mathbf{r}) - \boldsymbol{\pi})$ for $\mathbf{Q} \in NSD_2$ [Lindsay et al., 2008].

Lastly, we need the following assumption on the metric.

**Assumption 2.** *The gradient of $\phi$ at the trivial rate $\mathbf{o}$ is non-zero, i.e., $\nabla \phi^{\text{quad}}(\mathbf{r})|_{\mathbf{r}=\mathbf{o}} = \mathbf{a} + \mathbf{B}\mathbf{o} \neq 0$.*

The non-zero gradient assumption is reasonable for a concave $\phi^{\text{quad}}$, where it merely implies that the optimal classifier for the metric is not the uniform random classifier.

## 3 QUADRATIC METRIC ELICITATION

We now present our procedure for Quadratic Performance Metric Elicitation (QPME). We assume that the oracle's unknown metric is quadratic (Definition 3) and seek to estimate its parameters $(\mathbf{a}, \mathbf{B})$ by posing queries to the oracle. Unlike LPME, a simple binary search based procedure cannot be directly applied to elicit these parameters. Our approach instead approximates the quadratic metric by a linear function at a few select but *feasible* rate vectors and invokes LPME to estimate the local-linear approximations' slopes. One of the key challenges is to pick a small number of *feasible* rates for performing the local approximations and to reconstruct the original metric *just* from the estimated local slopes.

### 3.1 LOCAL LINEAR APPROXIMATION

We will find it convenient to work with a shifted version of the quadratic metric, centered at the point $\mathbf{o}$, the uniform random rate vector (see Proposition 1):

$$\phi^{\text{quad}}(\mathbf{r}; \mathbf{a}, \mathbf{B}) = \langle \mathbf{d}, \mathbf{r} - \mathbf{o} \rangle + \frac{1}{2}(\mathbf{r} - \mathbf{o})^T \mathbf{B}(\mathbf{r} - \mathbf{o}) + c$$
$$= \overline{\phi}(\mathbf{r}; \mathbf{d}, \mathbf{B}) + c, \quad (4)$$

where $\mathbf{d} = \mathbf{a} + \mathbf{B}\mathbf{o}$ and $c$ is a constant independent of $\mathbf{r}$, and so the oracle can be equivalently seen as responding with the shifted metric $\overline{\phi}(\mathbf{r}; \mathbf{d}, \mathbf{B})$.

Note that, due to the scale invariance condition in Definition 3, the largest singular value of $\mathbf{B}$ is bounded by 1. This is because $\|\mathbf{B}\|_2 \leq \|\mathbf{B}\|_F \leq 1$. Thus the metric $\phi^{\text{quad}}$ is 1-smooth and implies that it is locally linear around a given rate. To this end, let $z$ be a fixed point in $\mathcal{R}$, then the metric can be closely approximated by its first-order Taylor expansion in a small neighborhood around $\mathbf{z}$, for a constant $c'$ as follows:

$$\overline{\phi}(\mathbf{r}; \mathbf{d}, \mathbf{B}) \approx \langle \mathbf{d} + \mathbf{B}(\mathbf{z} - \mathbf{o}), \mathbf{r} \rangle + c'. \quad (5)$$

So if we apply LPME to the metric $\overline{\phi}$ with the queries $(\mathbf{r}_1, \mathbf{r}_2)$ to the oracle restricted to a small ball around $\mathbf{z}$, the procedure effectively estimates the slope of the vector $\mathbf{d} + \mathbf{B}(\mathbf{z} - \mathbf{o})$ in the above linear function (up to a small approximation error).

We exploit this idea by applying LPME to small neighborhoods around selected points to elicit the coefficients $\mathbf{a}$ and $\mathbf{B}$ for the original metric in (3). For simplicity, we will assume that the oracle is noise-free and later show robustness to noise and the query complexity guarantees in Section 5.

## 3.2 ELICITING METRIC COEFFICIENTS

We outline the main steps of Algorithm 1 below. Please see Appendix C for the full derivation.

**Estimate coefficients d (Line 1).** We first wish to estimate the linear portion $\mathbf{d}$ of the metric $\overline{\phi}$ in (4). For this, we apply the LPME subroutine to a small ball $\mathcal{S}_{\mathbf{o}} \subset \mathcal{S}$ of radius $\varrho < \rho$ around the point $\mathbf{o}$ (Fig. 2(a) illustrates this). Within this ball, the metric $\overline{\phi}$ approximately equals the linear function $\langle \mathbf{d}, \mathbf{r} \rangle + c'$ (see (5)), and so the LPME gives us an estimate of the slope of $\mathbf{d}$. From Remark 2, the estimates $\mathbf{f}_0 = (f_{10}, \ldots, f_{k0})$ approximately satisfy the following $(k-1)$ equations:

$$\frac{d_i}{d_1} = \frac{f_{i0}}{f_{10}} \qquad \forall\, i \in \{2, \ldots, k\}. \tag{6}$$

**Estimate coefficients B (Lines 2–4).** Next, we wish to estimate each column of the matrix $\mathbf{B}$ of the metric $\overline{\phi}$ in (4). For this, we apply LPME to small neighborhoods around points in the direction of standard basis vectors $\boldsymbol{\alpha}_j \in \mathbb{R}^k$, $j = 1, \ldots, k$. Note that within a small ball around $\mathbf{o} + \boldsymbol{\alpha}_j$, the metric $\overline{\phi}$ is approximately the linear function $\langle \mathbf{d} + \mathbf{B}_{:,j}, \mathbf{r} \rangle + c'$, and so the LPME procedure when applied to this region will give us an estimate of the slope of $\mathbf{d} + \mathbf{B}_{:,j}$. However, to ensure that the center point we choose is a feasible rate, we will have to re-scale the standard basis, and apply the subroutine to balls $\mathcal{S}_{\mathbf{z}_j}$ of radius $\varrho < \rho$ centered at $\mathbf{z}_j = \mathbf{o} + (\rho - \varrho)\boldsymbol{\alpha}_j$. See Figure 2(a) for the visual intuition. The returned estimates $\mathbf{f}_j = (f_{1j}, \ldots, f_{kj})$ approximately satisfy:

$$\frac{d_i + (\rho - \varrho)B_{ij}}{d_1 + (\rho - \varrho)B_{1j}} = \frac{f_{ij}}{f_{1j}} \quad \forall\, i \in \{2, \ldots, k\},\ j \leq i. \tag{7}$$

Now note that since we are only eliciting slopes using LPME, we always lose out on one degree of freedom. However, the matrix $\mathbf{B}$ is symmetric, thus we have $k(k+1)/2 - 1$ equations. There are $k(k+1)/2 + k$ unknown entities in $\mathbf{a}$ and $\mathbf{B}$, and to estimate them we need 1 more equation besides the normalization condition. For this, we apply LPME to a sphere $\mathcal{S}_{-\mathbf{z}_1}$ of radius $\varrho$ around rate $-\mathbf{z}_1$ as shown in Figure 2(a). The returned slopes $\mathbf{f}_1^- = (f_{11}^-, \ldots, f_{k1}^-)$ approximately satisfy:

$$\frac{d_2 - (\rho - \varrho)B_{21}}{d_1 - (\rho - \varrho)B_{11}} = \frac{f_{21}^-}{f_{11}^-}. \tag{8}$$

**Putting it together (Line 5).** By combining (6), (7) and (8), and denoting $F_{i,j,l} = f_{il}/f_{jl}$ and $F_{i,j,l}^- = f_{il}^-/f_{jl}^-$, we express each entry of $\mathbf{B}$ in terms of $d_1$ as follows:

$$B_{ij} = \Big( F_{i,1,j}(1 + F_{j,1,1}) - F_{i,1,j}F_{j,1,0}d_1 - F_{i,1,0}$$
$$+ F_{i,1,j}\frac{F_{2,1,1}^- + F_{2,1,1} - 2F_{2,1,0}}{F_{2,1,1}^- - F_{2,1,1}} \Big) d_1. \tag{9}$$

Using $\mathbf{d} = \mathbf{a} + \mathbf{B}\mathbf{o}$ and the fact that the coefficients are normalized, i.e., $\|\mathbf{a}\|_2^2 + \|\mathbf{B}\|_F^2 = 1$, we can obtain estimates

---

> **Algorithm 1: QPM Elicitation**
>
> **Input:** $\mathcal{S}$, Search tolerance $\epsilon > 0$, Oracle $\Omega$ with metric $\overline{\phi}$
> 1: $\mathbf{f}_0 \leftarrow \mathrm{LPME}(\mathcal{S}_{\mathbf{o}}, \epsilon, \Omega)$ with $\mathcal{S}_{\mathbf{o}} \subset \mathcal{S}$ and obtain (6)
> 2: **For** $j \in \{1, 2, \ldots, k\}$ **do**
> 3: $\quad \mathbf{f}_j \leftarrow \mathrm{LPME}\big(\mathcal{S}_{\mathbf{z}_j}, \epsilon, \Omega\big)$ with $\mathcal{S}_{\mathbf{z}_j} \subset \mathcal{S}$ and obtain (7)
> 4: $\mathbf{f}_1^- \leftarrow \mathrm{LPME}(\mathcal{S}_{-\mathbf{z}_1}, \epsilon, \Omega)$ with $\mathcal{S}_{-\mathbf{z}_1} \subset \mathcal{S}$ and obtain (8)
> 5: $\hat{\mathbf{a}}, \hat{\mathbf{B}} \leftarrow$ normalized solution derived from (9)
> **Output:** $\hat{\mathbf{a}}, \hat{\mathbf{B}}$

for $\mathbf{B}$ and $\mathbf{a}$ independent of $d_1$. Note that the derivation so far assumes $d_1 \neq 0$. This is based on Assumption 2 that at least one coordinate of $\mathbf{d}$ is non-zero, which w.l.o.g. we take to be $d_1$. In practice, we can identify a non-zero coordinate using $q$ trivial queries of the form $(\varrho\boldsymbol{\alpha}_i + \mathbf{o}, \mathbf{o}), \forall i \in [k]$.

**Technical novelty.** We emphasize that a key difference from Hiranandani et al. [2019a,b] is that they rely on a boundary point characterization which may not hold for general nonlinear metrics. Instead, we use structural properties of the metric to estimate local-linear approximations. While this may be a convenient approach (given LPME), as discussed in Section 1, implementing it involves non-trivial challenges, such as: (a) working with *only* slopes for the local-linear functions, (b) ensuring that the center points for the approximations are feasible, and (c) handling multiplicative errors in the analysis (see Section 5).

## 4 ELICITING FAIRNESS METRICS

Having understood the QPME procedure, we now discuss how our proposal can be applied to *quadratic metric elicitation for algorithmic fairness*. Like Hiranandani et al. [2020], we consider eliciting a metric that trades-off between predictive performance and fairness violation [Kamishima et al., 2012, Chouldechova, 2017, Menon and Williamson, 2018]. However, unlike Hiranandani et al. [2020], we handle general quadratic fairness violations and show how QPME can be easily employed to elicit group-fair metrics.

### 4.1 FAIRNESS PRELIMINARIES

The fairness setting is the same as the one in Section 3 except that we additionally have $m$ groups in the data and use $g \in [m]$ to denote the group membership. The groups are assumed to be disjoint, fixed, and known apriori [Agarwal et al., 2018]. We will work with a separate (randomized) classifiers $h^g : \mathcal{X} \to \Delta_k$ for each group $g$, and use $\mathcal{H}^g = \{h^g : \mathcal{X} \to \Delta_k\}$ to denote the set of all classifiers for $g$.

*Group predictive rates:* Similar to (1), we denote the group-conditional rates for $h^g$ by $\mathbf{r}^g(h^g, \mathbb{P}) \in \mathbb{R}^k$, where the $i$-th entry is additionally conditioned on group $g$:

$$r_i^g(h^g, \mathbb{P}) := \mathbb{P}(h^g = i | Y = i, G = g)\, \forall\, i \in [k]. \tag{10}$$

Analogous to the general setup, we denote the set of feasible rates for group $g$ by $\mathcal{R}^g = \{\mathbf{r}^g(h^g, \mathbb{P}) : h^g \in \mathcal{H}^g\}$.

**Example 3** (Fairness violation). A popular criterion for group fairness is the equal opportunity criterion of Hardt et al. [2016], which for a binary classification setup with $m$ protected groups, would require that $r_1^u = r_1^v$ for each pair of groups $(u, v)$. This can be formulated as constraints $|r_1^u - r_1^v| \leq \epsilon$, for some slack $\epsilon$ for all pairs $(u, v)$ [Agarwal et al., 2018], or more generally as a regularization term in the learning objective [Bechavod and Ligett, 2017, Hardt et al., 2016], by measuring the squared difference between the group rates: $\phi^{\text{EOpp}}((\mathbf{r}^1, \dots, \mathbf{r}^m)) = \binom{m}{2}^{-1} \sum_{v>u} (r_1^u - r_1^v)^2$. Another popular criterion is equalized odds, which requires equal rates across different protected groups [Bechavod and Ligett, 2017]. This again can be specified as a quadratic objective: $\phi^{\text{EO}}((\mathbf{r}^1, \dots, \mathbf{r}^m)) = [k\binom{m}{2}]^{-1} \sum_{v>u} \sum_{i=1}^k (r_i^u - r_i^v)^2$. Other fairness criteria that can be expressed as quadratic metrics include balance for the negative class, which for a binary classification problem is given by $\phi^{\text{BN}}((\mathbf{r}^1, \dots, \mathbf{r}^m)) = \binom{m}{2}^{-1} \sum_{v>u} (r_2^u - r_2^v)^2$ [Kleinberg et al., 2017], and the error-rate balance $\phi^{\text{EB}}((\mathbf{r}^1, \dots, \mathbf{r}^m)) = \binom{m}{2}^{-1} \frac{1}{2} \sum_{v>u} (r_1^u - r_1^v)^2 + (r_2^u - r_2^v)^2$ [Chouldechova, 2017] and their weighted variants.

In the next section, we introduce a general family of metrics that trades-off between an error term and a quadratic fairness violation term, for which we will need to define the rates for the overall classifier.

*Rates for overall classifier:* We construct the overall classifier $h : (\mathcal{X}, [m]) \rightarrow \Delta_k$ by predicting with classifier $h^g$ for group $g$, i.e. $h(\mathbf{x}, g) := h^g(\mathbf{x})$. We will be interested in both the fairness violation and predictive performance of the overall classifier. For the former, we will need the $m$ group-specific rates, represented together as a tuple:

$$\mathbf{r}^{1:m} := (\mathbf{r}^1, \dots, \mathbf{r}^m) \in \mathcal{R}^1 \times \cdots \times \mathcal{R}^m =: \mathcal{R}^{1:m}.$$

For the latter, we will measure the overall rates for $h$ as described in (1). The overall rates can also be written in terms of group-specific rates as: $\mathbf{r} = \sum_{g=1}^m \boldsymbol{\tau}^g \odot \mathbf{r}^g$, where $\boldsymbol{\tau}^g$ is just a constant vector whose $i$-th entry denote the prevalence of group $g$ within class $i$, i.e., $\mathbb{P}(G = g | Y = i)$.

## 4.2 FAIR QUADRATIC METRIC ELICITATION

We seek to elicit a metric that trades-off between predictive performance (a linear function of overall rates $\mathbf{r}$) and fairness violation (a quadratic function of group rates $\mathbf{r}^{1:m}$). For simplicity, we will denote the fairness metric in cost form, i.e., lower values are better.

**Definition 4.** (Fair Quadratic Performance Metric) *For misclassification costs* $\mathbf{a} \in \mathbb{R}^k$, $\mathbf{a} \geq 0$, *fairness violation costs* $\mathbb{B} = \{\mathbf{B}^{uv} \in PSD_k\}_{u,v=1,v>u}^m$, *and a trade-off parameter* $\lambda \in [0, 1]$, *we define:*

$$\phi^{\text{fair}}(\mathbf{r}^{1:m}; \mathbf{a}, \mathbb{B}, \lambda) := (1 - \lambda)\langle \mathbf{a}, \mathbf{1} - \mathbf{r} \rangle +$$
$$\lambda \frac{1}{2} \left( \sum_{v>u} (\mathbf{r}^u - \mathbf{r}^v)^T \mathbf{B}^{uv} (\mathbf{r}^u - \mathbf{r}^v) \right), \quad (11)$$

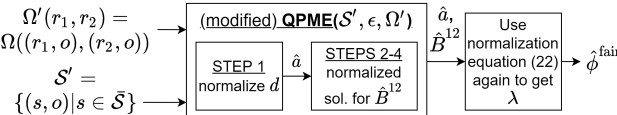

Figure 3: Eliciting Fair Quadratic Metrics for two groups. We formulate a $k$-dimensional elicitation problem and use a variant of QPME (Algorithm 1).

*where w.l.o.g. the parameters* $\mathbf{a}$ *and* $\mathbf{B}^{uv}$'s *are normalized:* $\|\mathbf{a}\|_2 = 1$, $\frac{1}{2} \sum_{v>u}^m \|\mathbf{B}^{uv}\|_F = 1$.

The coefficients $\mathbf{a}, \mathbf{B}^{uv}$'s are separately normalized so that the predictive performance and fairness violation are in the same scale, and we can additionally elicit the trade-off parameter $\lambda$. Analogous to Definitions 1–2, the problem of *Fair Quadratic Metric Elicitation* is as follows: given access to pairwise oracle queries of the form $\Omega(\hat{\mathbf{r}}_1^{1:m}, \hat{\mathbf{r}}_2^{1:m})$, recover a metric $\hat{\phi}^{\text{fair}} = (\hat{\mathbf{a}}, \hat{\mathbb{B}}, \hat{\lambda})$ such that $\|\phi^{\text{fair}} - \hat{\phi}^{\text{fair}}\| < \kappa$ under a suitable norm $\| \cdot \|$ for small $\kappa > 0$.

Similar to Section 2.2, we study the space of feasible rates $\mathcal{R}^{1:m}$ under the following mild assumption.

**Assumption 3.** *For all* $g \in [m]$, *the conditional distributions* $\mathbb{P}(Y = j | X, G = g)$, $j \in [k]$, *are distinct, i.e., there is some signal for non-trivial classification for each group.*

**Proposition 2** (Geometry of $\mathcal{R}^{1:m}$; Figure 2(b)). *For each group* $g$, *a classifier that predicts class* $i$ *on all inputs results in the same rate vector* $\mathbf{e}_i$. *The rate space* $\mathcal{R}^g$ *for each group* $g$ *is convex and so is the intersection* $\mathcal{R}^1 \cap \cdots \cap \mathcal{R}^m$, *which also contains the rate profile* $\mathbf{o} = \frac{1}{k} \sum_{i=1}^k \mathbf{e}_i$ *(achieved by the uniform random classifier) in the interior.*

**Remark 3** (Existence of sphere $\overline{\mathcal{S}}$). *There exists a sphere* $\overline{\mathcal{S}} \subset \mathcal{R}^1 \cap \cdots \cap \mathcal{R}^m$ *of radius* $\rho$ *centered at* $\mathbf{o}$. *Thus, a rate* $\mathbf{s} \in \overline{\mathcal{S}}$ *is feasible for each of the* $m$ *groups, i.e.,* $\mathbf{s}$ *is achievable by some classifier* $h^g$ *for each group* $g \in [m]$.

Because we allow a separate classifier for each group, Remark 3 implies that any rate $\mathbf{r}^{1:m} = (\mathbf{s}^1, \dots, \mathbf{s}^m)$ for arbitrary points $\mathbf{s}^1, \dots, \mathbf{s}^m \in \overline{\mathcal{S}}$ is achievable for some choice of group-specific classifiers $h^1, \dots, h^m$. This observation will be key to the elicitation algorithm we describe next.

## 4.3 ELICITING METRIC PARAMETERS $(\mathbf{a}, \mathbb{B}, \lambda)$

We present a strategy for eliciting fair metrics by adapting the QPME algorithm. For simplicity, we focus on the $m = 2$ case and extend our approach for $m > 2$ in Appendix D.

Observe that for a rate profile $\mathbf{r}^{1:2} = (\mathbf{s}, \mathbf{o})$, where the first group is assigned an arbitrary point in $\overline{\mathcal{S}}$ and the second group is assigned the uniform random classifier's rate $\mathbf{o}$, the

fair metric (11) becomes: $\phi^{\text{fair}}((\mathbf{s}, \mathbf{o}); \mathbf{a}, \mathbf{B}^{12}, \lambda)$

$$:= (1 - \lambda)\langle \mathbf{a}, \mathbf{1} - (\boldsymbol{\tau}^1 \odot \mathbf{s} + \boldsymbol{\tau}^2 \odot \mathbf{o})\rangle \ +$$
$$\frac{\lambda}{2}(\mathbf{s} - \mathbf{o})^T \mathbf{B}^{12}(\mathbf{s} - \mathbf{o})$$
$$:= \langle \mathbf{d}, \mathbf{s} - \mathbf{o}\rangle + \frac{1}{2}(\mathbf{s} - \mathbf{o})^T \mathbf{B}(\mathbf{s} - \mathbf{o})$$
$$:= \overline{\phi}(\mathbf{s}; \mathbf{d}, \mathbf{B}), \tag{12}$$

where $\mathbf{d} = -(1 - \lambda)\boldsymbol{\tau}^1 \odot \mathbf{a}$ and $\mathbf{B} = \lambda \mathbf{B}^{12}$, and we use $\boldsymbol{\tau}^1 + \boldsymbol{\tau}^2 = \mathbf{1}$ (the vector of ones) for the second step. The metric $\overline{\phi}$ above is a particular instance of the quadratic metric in (4). We can thus apply a slight variant of the QPME procedure in Algorithm 1 to solve the quadratic metric elicitation problem over the sphere $\mathcal{S}' = \{(\mathbf{s}, \mathbf{o}) \mid \mathbf{s} \in \overline{\mathcal{S}}\}$ with the modified oracle $\Omega'(\mathbf{r}_1, \mathbf{r}_2) = \Omega((\mathbf{r}_1, \mathbf{o}), (\mathbf{r}_2, \mathbf{o}))$.

The only change needed for the algorithm is in line 5, where we need to account for the changed relationship between $\mathbf{d}$ and $\mathbf{a}$ and need to separately (not jointly) normalize the linear and quadratic coefficients. With this change, the output of the algorithm directly gives us the required estimates. Specifically, from step 1 of Algorithm 1 and (6), we have $\hat{d}_i = -(1 - \lambda)\tau_i^1 \hat{a}_i$. By normalizing $\mathbf{d}$, we get $\hat{\mathbf{a}} = \frac{\mathbf{d}}{\|\mathbf{d}\|}$ for the linear coefficients. Similarly, steps 2-4 of Algorithm 1 and (9) allow us to express $\hat{B}_{ij} = \lambda \hat{B}_{ij}^{12}$ in terms of $\hat{a}_1$. After normalizing we directly get estimates $\hat{\mathbf{B}}^{12} = \hat{\mathbf{B}}/\|\hat{\mathbf{B}}\|_F$ for the quadratic coefficients.

Finally, because the linear and quadratic coefficients are separately normalized, the estimates $\hat{\mathbf{a}}$, $\hat{\mathbf{B}}^{12}$ are independent of the trade-off parameter $\lambda$. Given estimates $\hat{B}_{ij}^{12}$ and $\hat{a}_1$, we can now additionally estimate the trade-off parameter $\hat{\lambda}$. See Appendix D for details and Figure 3 for an illustration.

## 5 GUARANTEES

We discuss guarantees for the QPME procedure under the following practically relevant feedback model. The fair metric elicitation guarantees follow as a consequence.

**Definition 5** (Oracle Feedback Noise: $\epsilon_\Omega \geq 0$)**.** *Given rates $\mathbf{r}_1, \mathbf{r}_2$, the oracle responds correctly iff $|\phi^{\text{quad}}(\mathbf{r}_1) - \phi^{\text{quad}}(\mathbf{r}_2)| > \epsilon_\Omega$ and may be incorrect otherwise.*

In words, the oracle may respond incorrectly if the rates are close as measured by the metric. Since eliciting the metric involves offline computations of ratios, we make a regularity assumption ensuring that all components are well defined.

**Assumption 4.** *For the shifted quadratic metric $\overline{\phi}$ in (4), the gradients at the rate profiles $\mathbf{o}$, $-\mathbf{z}_1$, and $\{\mathbf{z}_1, \ldots, \mathbf{z}_q\}$, are non-zero vectors. Additionally, $\rho > \varrho \gg \epsilon_\Omega$.*

**Theorem 1.** *Given $\epsilon, \epsilon_\Omega \geq 0$, and a 1-Lipschitz metric $\phi^{\text{quad}}$ (Def. 3) parametrized by $\mathbf{a}, \mathbf{B}$, under Assumptions 1, 2, and 4, after $O\left(k^2 \log \frac{1}{\epsilon}\right)$ queries, Algorithm 1 returns a metric*

$\hat{\phi}^{\text{quad}} = (\hat{\mathbf{a}}, \hat{\mathbf{B}})$ *with* $\|\mathbf{a} - \hat{\mathbf{a}}\|_2 \leq O\left(\sqrt{k}(\epsilon + \sqrt{\varrho + \epsilon_\Omega/\varrho})\right)$ *and* $\|\mathbf{B} - \hat{\mathbf{B}}\|_F \leq O\left(k\sqrt{k}(\epsilon + \sqrt{\varrho + \epsilon_\Omega/\varrho})\right)$.

The proof of Theorem 1 uses the guarantee for LPME *only* as an intermediate step, and substantially builds on it to take into account the smoothness of the non-linear metric, the multiplicative errors in the slopes, and the feedback noise. We also provide a *finite sample version* of Theorem 1 in Corollary 1 (Appendix G), which states that the above result holds with high probability as long as (i) the hypothesis class of classifiers has finite capacity, and (ii) the number of samples used to estimate the rates is large enough.

**Theorem 2.** *(**Lower Bound**) For any $\epsilon > 0$, at least $\Omega(k^2 \log(1/(k\sqrt{k}\epsilon)))$ pairwise queries are needed to to elicit a quadratic metric (Def. 3) to an error tolerance of $k\sqrt{k}\epsilon$.*

Theorem 1 shows that the QPME procedure is robust to noise and its query complexity depends only *linearly* in the number of unknowns. Theorem 2 shows that the inherent complexity of the problem depends on the *number of unknowns*, thus our query complexity is optimal (barring the log term). So the $\tilde{O}(k^2)$ complexity is merely an artifact of our setup in Definition 3 being very general (with $O(k^2)$ unknowns). Indeed, with added structural assumptions on the metric, our proposal can be modified to considerably reduce the query complexity. For example, if we know that the matrix $\mathbf{B}$ is diagonal, then each LPME subroutine call needs to estimate only one parameter, which can be done with a constant number of queries, requiring a total of only $\tilde{O}(k)$ queries. We also stress that despite eliciting a more complex (non-linear) metric, the query complexity is still *linear in the number of unknowns*, which is same as prior linear elicitation methods [Hiranandani et al., 2019a,b].

## 6 EXPERIMENTS

We evaluate our approach on simulated oracles. Here we present results on a synthetically generated query space and in Appendix H.2 include results on real-world datasets.

**Eliciting quadratic metrics.** We first apply QPME (Algorithm 1) to elicit quadratic metrics in Definition 3. Like Hiranandani et al. [2020], we assume access to a $k$-dimensional sphere $\mathcal{S}$ centered at rate $\mathbf{o}$ with radius $\rho = 0.2$, from which we query rate vectors $\mathbf{r}$. The trends that we will discuss are robust to the sphere radius parameter $\rho$. Recall that in practice, Remark 1 guarantees the existence of such a sphere within the feasible region $\mathcal{R}$. We randomly generate quadratic metrics $\phi^{\text{quad}}$ parametrized by $(\mathbf{a}, \mathbf{B})$ and repeat the experiment over 100 trials for varying numbers of classes $k \in \{2, 3, 4, 5\}$. We run the QPME procedure with tolerance $\epsilon = 10^{-2}$. In Figures 4(a)–4(b), we show box plots of the $\ell_2$ (Frobenius) norm between the true and

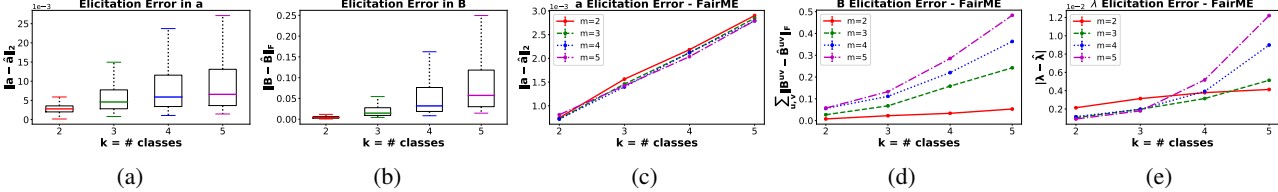

Figure 4: Average elicitation error over 100 metrics as a function of number of classes $k$ and groups $m$ for quadratic metrics in Definition 3 (a–b) and fairness metrics in Definition 4 (c–e). See Table 1 in Appendix H for the number of queries needed.

elicited linear (quadratic) coefficients. We generally find that QPME is able to elicit metrics close to the true ones. This holds for varying $k$, showing the effectiveness of our approach in handling multiple classes. The average number of queries we needed for elicitation over the 100 trials is provided in Table 1 in Appendix H. Note that the number of queries is $\tilde{O}(d)$ for eliciting a quadratic metric with $d = k^2$ unknowns, which clearly matches the lower bound in Theorem 2. See Appendix F for a discussion on the practicality of posing the requisite number of queries.

**Eliciting fairness metrics.** We next apply the elicitation procedure in Figure 3 with tolerance $\epsilon = 10^{-2}$ to elicit the fairness metrics in Definition 4. We randomly generate oracle metrics $\phi^{\text{fair}}$ parametrized by $(\mathbf{a}, \mathbb{B}, \lambda)$ and repeat the experiment over 100 trials and with varied number of classes and groups $k, m \in \{2, 3, 4, 5\}$. Figures 4(c)–4(e) show the mean elicitation errors for the the three parameters. For the linear predictive performance, the error $\|\mathbf{a} - \hat{\mathbf{a}}\|_2$ increases only with the number of classes $k$ and not groups $m$, as it is independent of the number of groups. For the quadratic violation term, the error $\sum_{u,v} \|\mathbf{B}^{uv} - \hat{\mathbf{B}}^{uv}\|_F$ increases with both $k$ and $m$. This is because the QPME procedure is run $\binom{m}{2}$ times for eliciting $\binom{m}{2}$ matrices $\{\mathbf{B}^{uv}\}_{v>u}$, and so the elicitation error accumulates with increasing $q$. Lastly, the elicited trade-off $\hat{\lambda}$ is seen to be close to the true $\lambda$ as well.

**Real-world datasets.** In App. H.2, we evaluate how well the elicited metric from QPME ranks a set of candidate classifiers trained on real-world datasets. We find that despite incurring elicitation errors, QPME achieves near-perfect ranking; whereas, the baseline metrics fail to do so.

## 7 RELATED WORK

Hiranandani et al. [2019a] formalized the problem of ME for binary classification with (quasi-)linear metrics and later extended it to the multiclass setting [Hiranandani et al., 2019b]. Unlike them, we elicit more complex quadratic metrics, and also provide an information-theoretic lower bound on the query complexity (Theorem 2). Prior works on ME offer no such lower bound guarantees. Learning linear functions passively using pairwise comparisons is a mature field [Joachims, 2002, Peyrard et al., 2017], but unlike their active learning counter-parts [Settles, 2009, Kane et al., 2017], these methods are not query-efficient. Studies such

as Qian et al. [2015] provide active linear elicitation strategies but with no guarantees and also work with a different query space. We are unaware of prior work that *provably* elicit a quadratic function, either passively or actively using pairwise comparisons. Our work is thus a significant first step towards active, nonlinear metric elicitation.

The use of metric elicitation for fairness is relatively new, with some work on eliciting *individual* fairness metrics [Ilvento, 2020, Mukherjee et al., 2020]. Hiranandani et al. [2020] is the only work we are aware of that elicits *group-fair* metrics, which we extend to handle more general metrics. Zhang et al. [2020] elicit the trade-off between accuracy and fairness using complex ratio queries. In contrast, we jointly elicit the predictive performance, fairness violation, and trade-off using simpler pairwise queries. Lastly, there has been work on learning fair classifiers under constraints [Zafar et al., 2017, Agarwal et al., 2018]. We take the regularization view of fairness, where the fairness violation is included in the objective [Kamishima et al., 2012].

Our work is also related to decision-theoretic *preference elicitation*, however, with the following key differences. We focus on estimating the utility function (metric) explicitly, whereas prior work such as [Boutilier et al., 2006, Benabbou et al., 2017] seek to find the optimal decision via minimizing the max-regret over a set of utilities. Studies that directly learn the utility [Perny et al., 2016] do not provide query complexity guarantees for pairwise comparisons. Formulations that consider a finite set of alternatives [Boutilier et al., 2006] are starkly different from ours, because the set of alternatives in our case (i.e. classifiers or rates) is infinite. Most of the papers in this literature focus on linear or bilinear [Perny et al., 2016] utilities except for [Braziunas, 2012] (GAI utilities) and [Benabbou et al., 2017] (Choquet integral); whereas, we focus on quadratic metrics which are useful for classification tasks, especially, fairness. We are not aware of any decision-theory literature that *provably* elicits quadratic (or polynomial) utility functions using pairwise comparisons.

Eliciting performance metrics bears similarities to *learning reward functions* in the inverse reinforcement learning literature [Wu et al., 2020, Abbeel and Ng, 2004, Levine et al., 2011, Sadigh et al., 2017] and the *Bradley-Terry-Luce model with features* in the learning-to-rank literature [Shah et al., 2015, Niranjan and Rajkumar, 2017]. However, in summary,

these studies focus on either eliciting linear utilities or passively learning utility functions. Our work is substantially different from them as we are tied to the geometry of the space of classification error statistics, and elicit quadratic utility functions using only pairwise comparisons, and particularly, in an active learning fashion. Moreover, we also provide query complexity bounds along with a lower bound. We further elaborate on the specific differences from these papers in Appendix I.

# 8 DISCUSSION

We have provided an efficient quadratic metric elicitation strategy with application to fairness, and with a query complexity that has the same dependence on the number of unknowns as that for linear metrics.

**Higher Order Polynomials:** We next show how our approach can be extended to elicit *higher-order polynomial* metrics. Thus our work not only increases the use-cases for ME but also opens the door for non-linear metric elicitation in other fields such as *active learning.*

Consider, e.g., a cubic polynomial:

$$\phi^{\text{cubic}}(\mathbf{r}) \coloneqq \sum_i a_i r_i + \frac{1}{2}\sum_{i,j} B_{ij} r_i r_j + \frac{1}{6}\sum_{i,j,l} C_{ijl} r_i r_j r_l,$$

where $\mathbf{B}$ and $\mathbf{C}$ are symmetric, and $\sum_i a_i^2 + \sum_{ij} B_{ij}^2 + \sum_{ijl} C_{ijl}^2 = 1$ (w.l.o.g., due to scale invariance). A quadratic approximation to this metric around a point $\mathbf{z}$ is given by:

$$\frac{1}{2}\left(\sum_{i,j} B_{ij} r_i r_j + \sum_{i,j,l} C_{ijl}(r_i - z_i)(r_j - z_j) z_l\right) + \sum_i a_i r_i + c,$$

where $c$ is a constant not affecting the oracle responses. We can estimate the parameters of this approximation by applying the QPME procedure from Algorithm 1 with the metric centered at an appropriate point, and its queries restricted to a small neighborhood around $\mathbf{z}$. Running QPME once using a sphere around the point $\mathbf{z}_l = \mathbf{o} + (\varrho - \varrho')\boldsymbol{\alpha}_l$, where $\varrho' < \varrho$ will elicit one face of the tensor $\mathbf{C}_{[:,:,l]}$ upto a scaling factor. Thus, it will require us to run the QPME procedure $k$ times around the basis points $\mathbf{z}_l = \mathbf{o} + (\varrho - \varrho')\boldsymbol{\alpha}_l \ \forall l \in [k]$. Since we elicit scale-invariant quadratic approximation, we would need additional run of QPME procedure around the point $\mathcal{S}_{-\mathbf{z}_1}$ to elicit all the coefficients. Thus, we can recover the metric $\hat{\phi}^{\text{cubic}} = (\hat{\mathbf{a}}, \hat{\mathbf{B}}, \hat{\mathbf{C}})$ with as many queries as the number of unknowns, i.e, $\tilde{O}(k^3)$ in the cubic case.

For a $d$-th order polynomial, one can recursively apply this procedure to estimate $(d-1)$-th order approximations at multiple points, and similarly derive the polynomial coefficients from the estimated local approximations.

**Handling large number of classes:** For applications where $k$ is very large, the parameterization discussed in Section 2 may not be applicable in its current form. For example, when $k = 1000$, the quadratic metric in (3) would use $O(1000^2)$ parameters, an exorbitantly high number to elicit in practice. Note that the presence of $k^2$ unknowns is an artifact of the problem formulation, and *not* of our proposed procedure. Moreover, as shown in Theorem 2, it is *theoretically impossible* to estimate $k^2$ unknowns with fewer than $\tilde{O}(k^2)$ queries. While our QPME procedure does indeed match this lower bound, in practice, we do not expect it to be applied to estimate such an over-parameterized metric. Instead, for such large-scale settings, we recommend making reasonable assumptions on the metric to reduce the number of unknowns, e.g., by having multiple classes share the same parameter, and the query complexity of QPME would then only depend *linearly* on the reduced number of unknowns. For instance, in Table 4 (Appendix H.2), we show that by simplifying the metric with structural assumptions, one can use fewer queries in practice to get comparable results.

**Advantages:** Our proposal comes with many practical advantages: (a) *Fairness:* we are aware of no prior work that can elicit fair quadratic metrics, particularly with provable guarantees; (b) *Transportability:* our method is independent of the population $\mathbb{P}$, which allows any metric that is elicited using one dataset or model class to be applied to other applications, as long as the expert believes the tradeoffs to be the same; and (c) *Feasibility:* we ensure that the rates are feasible throughout the elicitation (i.e., are achievable by classifiers), which allows the flexibility to deploy systems that either compare classifiers or compare rates.

**Limitations:** Limitations of our work include the assumption that the metric has a parametric form, which can be restrictive in some cases, and not providing a concrete answer to who the oracles should be. One should also be cautious in applying ME to eliciting fairness metrics, as failures here could exacerbate the adverse effects on protected groups.

**Future Work:** To thoroughly answer the above questions, we are actively conducting user studies on collecting preference feedback using intuitive visualizations of rates [Shen et al., 2020, Beauxis-Aussalet and Hardman, 2014] or classifiers [Ribeiro et al., 2016] . Please see Appendix J to take a peek into the future work, where we discuss findings from a preliminary user study.

## Acknowledgements

This research was funded by Google Research. The authors would like to thank Safinah Ali, Sohini Upadhyay, and Elena Glassman for helping with the pilot user study discussed in Appendix J.

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
