# OpenReview forum: "Quadratic Metric Elicitation for Fairness and Beyond"
_auai.org/UAI/2022/Conference — UAI 2022 Oral_

### Official Review · Reviewer_bTFu · 2022-04-11

**Q2(1) Originality/Novelty:** 3
**Q2(2) Significance/Impact:** 3
**Q2(3) Correctness/Technical Quality:** 3
**Q2(6) Clarity Of Writing:** 3
**Q6 Overall Score:** 7
**Q8 Confidence In Your Score:** 2

**Q1 Summary And Contributions:**

The authors provide an algorithm that elicits a quadratic metric based on queries to the decision maker. This extends a recent paper where linear metrics were elicited. The authors show that their algorithm is useful for eliciting group-fair metrics. The algorithm and experimental results are supplemented with theoretical ones.

**Q2 Assessment Of The Paper:**

More detailed information regarding each of these aspects is given below:

**Q2(4) Quality Of Experiments (Optional):**

3: Good: The experimental evaluation is adequate, and the results convincingly support the main claims.

**Q2(5) Reproducibility:**

3: Good: Key resources (e.g., proofs, code, data) are available and key details (e.g., proofs, experimental setup) are sufficiently well-described for competent researchers to confidently reproduce the main results.

**Q3 Main Strengths:**

I have to say that I am not very familiar with the topic of the paper. Reading the introduction was quite challenging for me, since it is presented on a fairly abstract level. Yet, when I saw Section 2 and the following discussions, I was still able to understand the main ideas and the reason why this work is being done. I see it as a big strenght of the paper that the authors can convey their message and ideas in the technical part even to someone who did not follow the introduction.

I can imagine settings where solving the studied problem is relevant and important.

**Q4 Main Weakness:**

I think that the introduction would be much more appealing to people like me, who are not directly in the area of the paper, if it presented a very concrete example where the paper's results would be useful (on the other hand, it is not apriori clear that appealing to such readers should be the main goal; yet, this way the authors could improve the impact of their work).

**Q5 Detailed Comments To The Authors:**

Minor comments:

In Section 2, I found the  P symbol (as in P(X,Y)) to be somewhat confusing. It seems to mean both a general way to express "probability" and a distribution that the authors discuss.

"Our proposal easily extends" <-- at this point it is really not clear what "Our proposal" refers to.

**Q7 Justification For Your Score:**

I am not familiar with this line of research and I cannot evaluate it very well. Yet, the authors convinced me that they achieved something useful.

**Q9 Complying With Reviewing Instructions:**

1: Yes.

---

### Official Review · Reviewer_aaJk · 2022-04-13

**Q2(1) Originality/Novelty:** 3
**Q2(2) Significance/Impact:** 3
**Q2(3) Correctness/Technical Quality:** 3
**Q2(6) Clarity Of Writing:** 4
**Q6 Overall Score:** 9
**Q8 Confidence In Your Score:** 4

**Q1 Summary And Contributions:**

The paper extends the framework of *metric elicitation* for classification tasks to the class of quadratic metrics, and even polynomial ones.
It proceeds by using an algorithm designed for linear metric elicitation to estimate slopes of gradients at feasible points and carefully reconstructing the metric so as to monitor error propagation, thus deriving guarantees wrt noise. The approach is convincingly illustrated on fairness problems.

**Q2 Assessment Of The Paper:**

More detailed information regarding each of these aspects is given below:

**Q2(4) Quality Of Experiments (Optional):**

3: Good: The experimental evaluation is adequate, and the results convincingly support the main claims.

**Q2(5) Reproducibility:**

3: Good: Key resources (e.g., proofs, code, data) are available and key details (e.g., proofs, experimental setup) are sufficiently well-described for competent researchers to confidently reproduce the main results.

**Q3 Main Strengths:**

The paper presents non-trivial advances to a subject that is difficult and important. The research question is clearly stated, clearly motivated, and answered with algorithmic, theoretical and empirical arguments that amount to a well-rounded perspective. In particular, as metric elicitation is a subfield of active preference learning, it is nice to provide a framework able to address non-linear preferences, while proposing a (nearly) minimal number of queries, ensuring these queries are meaningful by restricting them to questions about feasible alternatives, and being robust to noise.

**Q4 Main Weakness:**

While the paper is impressive, I have two quibbles:
1. I am not sure it belongs to a conference anymore. The paper is not really self-sufficient without the supplementary material, which I believe sums up the previous work of the authors. This shows the work is mature enough for submission (so that the reviewing process could be more thorough and constructive) and publication (so that a reader would find everything in the same place, narrated in the correct order) to a journal.
2. I believe the positioning wrt (active) preference elicitation / preference learning is a bit shallow. For instance, is using MLE as a subroutine a good way to learn preferences modeled with piecewise linear utility functions, as in the seminal work of Jacquet-Lagrèze/Siskos, or a Choquet integral like in Benabbou/Perny/Viappiani or Labreuche/Hullermeier?
Also, it might be relevant to explore the connections with multiple objective optimization, where the notion of preference is also actively discovered.

**Q5 Detailed Comments To The Authors:**

None.

**Q7 Justification For Your Score:**

The paper is very strong in many aspects (see Q3 above). Its main flaw is being **too strong** for a conference, because it is yet-another-brick in a framework accruing non-trivial notions by the same authors.

**Q9 Complying With Reviewing Instructions:**

1: Yes.

---

### Official Review · Reviewer_6GnD · 2022-04-15

**Q2(1) Originality/Novelty:** 3
**Q2(2) Significance/Impact:** 3
**Q2(3) Correctness/Technical Quality:** 3
**Q2(6) Clarity Of Writing:** 4
**Q6 Overall Score:** 7
**Q8 Confidence In Your Score:** 3

**Q1 Summary And Contributions:**

The authors develop an LPME-based method to deal with metric elicitation problems when metrics could be quadratic. Under mild assumptions, they could provide a detailed theoretical analysis of their method. Applications to fairness metric elicitation are also provided.

**Q2 Assessment Of The Paper:**

More detailed information regarding each of these aspects is given below:

**Q2(4) Quality Of Experiments (Optional):**

3: Good: The experimental evaluation is adequate, and the results convincingly support the main claims.

**Q2(5) Reproducibility:**

3: Good: Key resources (e.g., proofs, code, data) are available and key details (e.g., proofs, experimental setup) are sufficiently well-described for competent researchers to confidently reproduce the main results.

**Q3 Main Strengths:**

1. The paper provides a sound method to deal with the quadratic metric elicitation problem, which is a non-trivial extension to previous linear cases.
2. Under mild assumptions, the paper develops an approximation bound on the parameters with a bounded amount of queries. A lower bound is also provided to demonstrate the complexity of the problem.
3. The paper is generally well-written with detailed analysis and examples.

**Q4 Main Weakness:**

I do not come up with any major weaknesses in the paper, while I have several questions as mentioned below.

**Q5 Detailed Comments To The Authors:**

1. How can we get the $\mathcal{S}_{\mathbf{o}}$ in practice? The methods need the prior knowledge of this set and I do not know to obtain it.
2. Should the lower bound on the number of queries in Theorem 2 mean $\Omega(k^2\log (1 / (k\sqrt{k}\epsilon)))$? I think the additional parentheses should be added.

**Q7 Justification For Your Score:**

As shown above, I think this paper provides a non-trivial method for an important method. Further theoretical analysis is also provided. As a result, I have a positive viewpoint of the paper.

**Q9 Complying With Reviewing Instructions:**

1: Yes.

---

### Official Review · Reviewer_hC46 · 2022-04-17

**Q2(1) Originality/Novelty:** 2
**Q2(2) Significance/Impact:** 2
**Q2(3) Correctness/Technical Quality:** 3
**Q2(6) Clarity Of Writing:** 2
**Q6 Overall Score:** 6
**Q8 Confidence In Your Score:** 4

**Q1 Summary And Contributions:**

The paper considers the metric elicitation problem and proposes an approach that can include metrics defined by quadratic functions of predictive rates. The paper also applies the proposed method to fairness audit problems and presents a potential extension to eliciting polynomial metrics.

**Q2 Assessment Of The Paper:**

More detailed information regarding each of these aspects is given below:

**Q2(4) Quality Of Experiments (Optional):**

2: Fair: The experimental evaluation is weak: important baselines are missing, or the results do not adequately support the main claims.

**Q2(5) Reproducibility:**

2: Fair: Key resources (e.g., proofs, code, data) are unavailable but key details (e.g., proof sketches, experimental setup) are sufficiently well-described for an expert to confidently reproduce the main results.

**Q3 Main Strengths:**

The strength of the paper is represented by an improvement in functional form complexity that a Metric Elicitation (ME) framework can handle (from linear/quasi-linear to quadratic). The paper approaches the problem by linearly approximating the quadratic function by multiple linear functions and combining the estimated slopes to reconstruct the original quadratic metric.

**Q4 Main Weakness:**

I have several questions/concerns regarding the connection between ME and fairness considerations (details can be found in "Detailed Comments to the Authors").

First, I am not sure if ME is a good way to frame fairness in ML research.

Second, it is not clear to me whether the problem setup (Section 2) is w.r.t. ME only, or, the problem setup has some fairness consideration built-in.

Third, I don't think the formulation of fairness violation as a quadratic function (of predictive rates) applies to the fairness notions listed in the paper.

**Q5 Detailed Comments To The Authors:**

Question 1: the connection between ME framework and fairness audits

In Figure 1 (adapted from Hiranandani et al., 2019a), there is an Oracle that contains performance metrics, and an elicitation procedure that elicits metrics using preference feedback. I am having some difficulties framing fairness inquiries in the ME framework. What is the Oracle in fairness audits? Why there is an elicitation procedure to "elicit" fairness violation metric? What about fairness notions that are not based on relative preference or predictive rates?

Question 2: the problem setup and motivation

In Section 2 where the problem setup is presented, it is unclear what the motivation is behind the setup. For example, in Equation 1, the paper defines the primary object of interest, the predictive rate functions. If I understand it correctly, the rates are not (explicit) functions of protected features and are not group-dependent. If so, how can the framework model fairness audits? It seems to me that the ME framework is a purely technical consideration and the fairness audits are just possible applications instead of the motivation.

Question 3: the formulation of fairness violation as quadratic functions

In Example 2, the paper listed several quadratic functions for fairness. I am hesitant to agree with the presented formulations. For example, Equalized Odds is first presented in Hardt et al., 2016, and there are no quadratic functions proposed in their paper to measure the violation. To the best of my knowledge, Equalized Odds is a conditional independence based fairness notion, whose weaker notion might be measured via predictive rates. A clearer citation would be very helpful to avoid potential misunderstandings.

---------------------
Post rebuttal update:

Please kindly consider including the proposed modifications of the manuscript to avoid potential misunderstandings. The evaluation score has been updated accordingly.

**Q7 Justification For Your Score:**

The paper is not clearly motivated and I am not sure about the contribution that this paper is aiming for. On the one hand, the majority of the paper deals with multi-class metric elicitation with quadratic functions of predictive rates, and the fairness seems to be a side note, which is only one way to apply the proposed method; on the other hand, the title seems to suggest that fairness consideration is the motivation of the proposed approach.

---
Post rebuttal update included
---

**Q9 Complying With Reviewing Instructions:**

1: Yes.

---

### Decision · Program_Chairs · 2022-05-15

**Decision:**

Accept (Oral)

**Comment:**

Meta Review: All reviewers agree that this is an interesting and valid contribution, following an quite interactive discussion between reviewers and authors.

Authors should include the modifications suggested by the discussions in the final paper.